# Overcoming HPV Vaccine Hesitancy in Japan: A Narrative Review of Safety Evidence, Risk Communication, and Policy Approaches

**DOI:** 10.3390/vaccines13060590

**Published:** 2025-05-30

**Authors:** Takayuki Takahashi, Megumi Ichimiya, Misa Tomono, Rio Minoura, Takahiro Kinoshita, Yousuke Imanishi, Masahiko Sakamoto, Makiko Mitsunami, Mihyon Song, Kanako Inaba, Daisuke Shigemi

**Affiliations:** 1Department of Obstetrics and Gynecology, Federation of National Public Service Personnel Mutual Aid Associations, Tachikawa Hospital, Tokyo 100-0013, Japan; 2Minpapi Association, Tokyo 100-8926, Japan; megphbs17@gmail.com (M.I.); misa_tomono@icloud.com (M.T.); riomino85@gmail.com (R.M.); ogmc.dmat.t.kinoshita@gmail.com (T.K.); rasheed0421@yahoo.co.jp (Y.I.); masahikoskmt77@yahoo.co.jp (M.S.); maki3273@gmail.com (M.M.); mihyon0123@gmail.com (M.S.); kaaaaaaana@gmail.com (K.I.); daishige2019@gmail.com (D.S.); 3Department of Prevention and Community Health, George Washington University, Washington, DC 20052, USA; 4Department of Pediatrics, Saku Central Hospital, Nagano 812-0041, Japan; 5Master of Medical Sciences in Clinical Investigation Program, Harvard Medical School, Boston, MA 02115, USA; 6Marunouchinomori Ladies Clinic, Tokyo 100-0005, Japan; 7Inaba Clinic, Tokyo 150-0043, Japan; 8Department of Obstetrics and Gynecology, Nippon Medical School, Tokyo 113-8602, Japan

**Keywords:** human papillomavirus, vaccine hesitancy, cervical cancer, safety evidence, risk communication, policy approaches, Japan

## Abstract

Human papillomavirus (HPV) infection remains a principal cause of cervical cancer worldwide. Although large-scale vaccination efforts have substantially lowered HPV infection rates and precancerous lesions, not all regions have achieved high coverage. In Japan, proactive HPV vaccine recommendations were suspended from 2013 to 2022 due to concerns over alleged adverse events, causing vaccination rates to drop from over 70% to below 1%. This narrative review synthesized research published from 2014 to 2025 in PubMed, Cochrane Library, and Google Scholar, focusing on English-language studies. Inclusion criteria encompassed analyses of HPV vaccine efficacy or safety, policies related to vaccination in Japan or other countries, and investigations into vaccine hesitancy or media influences. Data were categorized into five thematic areas: historical and policy contexts, evidence of vaccine safety and efficacy, societal drivers of hesitancy, communication strategies, and administrative or clinical interventions. Evidence robustly confirms the HPV vaccine’s favorable safety profile, with severe adverse events appearing exceedingly rare. Nonetheless, media sensationalism and limited risk communication in Japan perpetuated mistrust, impeding vaccination uptake. Comparisons with Denmark and Ireland indicate that transparent, interactive risk communication can restore coverage to near-pre-suspension levels. Japan’s recent policy reforms, including reinstating proactive recommendations and catch-up initiatives, have begun to reverse vaccination hesitancy. Sustained policy support, evidence-based messaging, and empathetic engagement with communities are central to rebuilding trust in the HPV vaccine. Lessons from best international practices emphasize the importance of multifaceted interventions, collaborative stakeholder engagement, and transparent risk communication to reduce the burden of HPV-related malignancies.

## 1. Introduction

Human papillomavirus (HPV), a prevalent sexually transmitted pathogen, plays a central role in the pathogenesis of several malignancies, most notably cervical cancer. According to the World Health Organization (WHO), approximately 604,000 new cases of cervical cancer were reported in 2020, highlighting a continuing global public health challenge for women’s health [1]. In particular, Japan reports approximately 13,000 new cervical cancer diagnoses annually, with roughly 4000 deaths attributed to the disease [2]. These figures underscore the urgency of implementing effective preventive measures, particularly among younger populations, where HPV-associated oncogenesis poses a serious concern.

Since 2006, an increasing body of evidence has demonstrated that HPV vaccination programs substantially reduce HPV infection rates and precancerous lesions among young women [3,4]. This trend is especially pronounced in regions such as Europe and Oceania, where high vaccine coverage has been maintained, positioning HPV vaccination as a cornerstone of primary prevention against cervical cancer [5]. However, in Japan, the Ministry of Health, Labour and Welfare suspended the proactive recommendation of the HPV vaccine in 2013, leading to a marked decline in vaccination coverage—from an initial rate exceeding 70% to below 1% within just a few months [6,7]. As a result, a significant cohort of young women, often referred to as “the lost generation”, missed the opportunity to be vaccinated during the optimal age window for immunization. Evidence suggests that extensive media reports and social media discussions of suspected post-vaccination adverse events contributed to widespread hesitancy [7,8,9]. Similar temporary drops in vaccine uptake were observed in Denmark and Ireland around the same period; however, both countries restored vaccination coverage within a relatively short timeframe through comprehensive information dissemination and interactive campaigns [10,11].

Ample research conducted since 2015, comprising large-scale international studies and surveillance data, indicates that the risk of severe adverse events associated with HPV vaccination remains comparable to other routine vaccines [12,13,14]. Public health authorities, including the U.S. Centers for Disease Control and Prevention (CDC) and the European Medicines Agency (EMA), have consequently emphasized the efficacy of HPV vaccines in preventing HPV-related malignancies while noting that adverse event profiles remain comparable to those of other routine immunizations [15,16,17]. Several large-scale investigations within Japan have similarly found no statistically significant correlation between the vaccine and chronic pain or neurological impairments [18,19]. Nevertheless, over eight years of effectively suspended vaccination efforts have culminated in a broader pool of unvaccinated cohorts and a concomitant erosion of public trust in vaccine safety [6,20]. Drawing on lessons learned from international examples, such as those in Denmark and Ireland [10,11,21,22], targeted risk communication strategies are becoming increasingly essential to counter misinformation and regain public confidence.

In this review, we examine the trajectory of HPV vaccine hesitancy in Japan, elucidate the social and historical factors underpinning the prolonged downturn in vaccination coverage, and synthesize global and domestic safety evidence. Furthermore, we discuss communication approaches and policy strategies aimed at reversing low uptake and ultimately reducing the long-term burden of preventable HPV-related diseases associated with this vaccination gap.

## 2. Materials and Methods

This article adopts a narrative review approach to comprehensively examine research addressing the decline in HPV vaccination coverage, communication strategies, safety evaluations, and policy interventions. The search strategy was designed to ensure transparency and minimize selection bias. Specifically, searches were conducted in three primary databases—PubMed, Cochrane Library, and Google Scholar—targeting English-language articles published between 2014 and March 2025. A total of 134 articles were identified across these databases. Google Scholar was specifically employed for broader, free-text searches to identify relevant studies that may not have been captured through standard keyword-based queries in other databases. While PubMed and Cochrane Library were prioritized for their rigorous peer-reviewed content and established indexing systems, Google Scholar was included to capture other studies that may not be indexed in traditional databases. The primary search terms included “HPV vaccine”, “cervical cancer”, “vaccine hesitancy”, “risk communication”, “immunization policy” and “catch-up vaccination”. Both titles and abstracts were initially screened for relevance. After the initial screening, two independent researchers assessed the full text of the identified studies for eligibility. Disagreements were resolved through consensus between the researchers.

Studies selected for inclusion met at least one of the following criteria:Original or review articles evaluating the efficacy or safety of HPV vaccines,Policy analyses focused on HPV vaccination in Japan or other countries,Investigations centered on vaccine hesitancy or risk communication,Research documenting trends in HPV vaccination coverage in Japan since 2014.

Exclusion criteria included studies that focused on unrelated age groups or disease domains, lacked robust empirical evidence, or were non-peer-reviewed publications. When conflicting studies were identified, researchers discussed the findings to reconcile discrepancies and selected the most methodologically sound evidence. This process ensured that only high-quality and relevant studies were included in the review.

The final set of articles was organized around five key perspectives: (1) the historical and policy context of HPV vaccination; (2) evidence on vaccine safety and efficacy; (3) societal factors such as media coverage and social media-based information sharing; (4) communication and educational interventions; and (5) administrative or clinical strategies and outcome metrics. Drawing on these lenses, the discussion aims to elucidate the factors contributing to low vaccination uptake and to explore multifaceted approaches for recovering coverage rates.

## 3. Japan’s HPV Vaccination Crisis

### 3.1. Historical Context and Rapid Decline

Public funding for HPV vaccination in Japan first became available at the municipal level around 2010. In April 2013, HPV vaccination was officially incorporated into the national vaccination schedule under the Immunization Act, covering girls in grades 6 through 10 (approximately ages 12–16) [6,23,24]. Early estimates suggest that vaccination rates exceeded 70%, indicating a relatively high coverage by international standards [6,25]. However, in June 2013, the Ministry of Health, Labour and Welfare decided to suspend proactive recommendations for HPV vaccination, which resulted in a precipitous decline in coverage. Within a year, first-dose uptake plummeted to below 1%, resulting in a substantial cohort of unvaccinated adolescents, sometimes referred to as the “lost generation” [7,20].

Initially, the ministry framed the suspension as a “temporary measure”, citing the need to investigate potential causal links between vaccination and suspected adverse events. Nonetheless, no clear policy shift emerged for an extended period, leading to persistently low vaccination rates over the long term [6,26]. Throughout this interval, numerous young women missed the opportunity to be immunized against HPV, prompting concerns regarding the future burden of cervical cancer that might otherwise have been preventable [20,27].

#### 3.1.1. From Public Subsidies to Official Recognition as a Routine Vaccine

Japan’s HPV vaccination program gained momentum following the vaccine’s approval in 2009, with municipal governments introducing public subsidies the following year [7]. Building on this foundation, the vaccine was elevated in April 2013 to “routine vaccination” status under the Immunization Act, accompanied by near-universal cost coverage and a legally defined “duty to make an effort” [23,25]. At that time, domestic and international evidence strongly supported the vaccine’s efficacy [3,4,5]. The rapid initial uptake demonstrated the feasibility of a large-scale, nationwide immunization platform for adolescent girls soon after the vaccine attained routine vaccination status.

#### 3.1.2. Impact of the Abrupt Decline in Coverage

The June 2013 suspension of proactive HPV vaccination recommendations triggered a dramatic fall in willingness to vaccinate. Within a few months, three-dose completion rates dropped to near zero among newly eligible cohorts, compared to the over 70% coverage levels observed in earlier cohorts [7,20]. This rapid collapse, widely regarded as “unprecedented” both within and outside Japan [6,10], led to persistently low vaccination rates and the emergence of numerous unvaccinated cohorts—thereby posing a significant public health concern [26,28].

### 3.2. Extended Suspension and Partial Resumption

When the Ministry of Health, Labour and Welfare halted its active recommendations in June 2013, it cited the need for a more thorough investigation of the suspected adverse effects. However, the suspension lasted for over eight years, during which time Japan’s HPV vaccination policy remained effectively inactive [6,26].

#### 3.2.1. Background Factors Contributing to Long-Term Suspension

Multiple factors appear to have perpetuated this protracted suspension. Media coverage and advocacy by individuals reporting adverse events contributed to heightened public concern, and concerns over potential legal action regarding vaccine-related harm also played a role [7,9]. Furthermore, a lack of consensus within the medical community and professional societies reinforced the Ministry’s cautious stance. While professional bodies such as the Japan Society of Obstetrics and Gynecology and the Japan Pediatric Society repeatedly called for the prompt reinstatement of vaccination recommendations, surveys indicate that frontline medical providers themselves often hesitated to recommend the vaccine during this period, due to ongoing uncertainties surrounding the reported adverse events [23,29,30]. In addition to these findings, Machida et al. (2023) conducted a large-scale cluster analysis among 3790 women from the catch-up vaccination generation in Japan [31]. Their study identified three distinct patterns of vaccine hesitancy—acceptance, neutrality, and refusal—linked with sociodemographic factors and media exposure. This nationwide data highlights the heterogeneity of vaccine attitudes beyond localized surveys and underscores the complexity of the vaccine hesitancy phenomenon in Japan.

#### 3.2.2. Reinstatement Measures Since 2022

In November 2021, the Ministry of Health, Labour and Welfare—citing evidence supporting vaccine safety and the implementation of new support systems—announced that proactive HPV vaccination recommendations would resume after an 8.5-year hiatus [26,32]. Beginning in April 2022, local authorities reissued vaccination coupons and reminders. At the same time, “catch-up vaccination” provisions were introduced for women born between 1997 and 2005, granting them three years of cost-free access to the vaccine [6,26]. Although initial reports suggest some increase in uptake, coverage rates generally remain below pre-suspension levels [27,33]. Persistent informational gaps, compounded by longstanding public skepticism, have hindered a rapid recovery in HPV vaccination rates, leaving many challenges unresolved. Complementing these factors, a recent population-based study by Oka et al. (2025) in Osaka City demonstrated significant disparities in HPV vaccination uptake related to neighborhood socioeconomic status and access to vaccination facilities [34]. Their findings suggest that structural barriers, including socioeconomic deprivation and limited healthcare access, contribute substantially to uneven vaccination coverage. Addressing these social determinants is thus essential for designing effective interventions to increase vaccine uptake across all communities.

### 3.3. Media Influence and Public Hesitancy

In Japan, the rapid spread of HPV vaccine hesitancy has been closely linked to media narratives and the proliferation of unverified information via social media channels.

#### 3.3.1. Intensive Media Coverage

Starting in early 2013, sensationalized stories on severe pain and motor disturbances allegedly linked to HPV vaccination began to dominate television broadcasts and newspaper coverage, often without clearly stating the absence of scientifically established causal relationships [7]. The suspension in June 2013 further amplified these reports, shifting media coverage toward “unknown risks” and relegating discussions of vaccine efficacy and international safety data to the background [9,35]. The resulting wave of public anxiety played a significant role in the collapse of HPV vaccination rates.

#### 3.3.2. Misinformation on Social Media and Heightened Public Anxiety

Following the suspension, social media platforms increasingly featured personal accounts of post-vaccination symptoms and facilitated the spread of unverified information [28,36]. Meanwhile, official guidance from health authorities remained sporadic, allowing negative perceptions to proliferate, particularly among adolescents and their parents. A social media analysis revealed that even after proactive recommendations were reinstated in 2021, a substantial share of user-generated content continued to express apprehensions about vaccine safety [20].

#### 3.3.3. Entrenched Vaccine Hesitancy

Vaccine hesitancy in Japan has been attributed not only to media sensationalism and the amplification of fears on social media, but also to a longstanding lack of public trust in immunization policy, insufficient risk communication by government authorities, and limited coordination among healthcare professionals [8,29,30]. Following the intensive media coverage and suspension in 2013, perceptions that “the HPV vaccine is potentially harmful” persisted despite subsequent scientific clarifications, contributing to continued low uptake [10,11]. In contrast to several Western countries that managed to reverse downward trends in HPV vaccination within a few years, Japan has experienced a protracted crisis.

Thus, media coverage, social media discourse, and cautious official policy reinforced one another, compounding Japan’s “vaccination crisis” and delaying efforts to restore public trust and expand access through catch-up initiatives. As discussed in subsequent sections, lessons gleaned from international best practices and evidence-based risk communication will be vital in guiding Japan’s recovery from historically low HPV vaccination rates.

## 4. HPV Vaccine Safety: Global and Domestic Evidence

### 4.1. Post-2015 International Data

Since its global introduction in 2006, HPV vaccination has been adopted by numerous countries, with safety data continuing to accumulate, particularly after 2015. A comprehensive systematic review synthesizing findings from randomized controlled trials (RCTs) and observational studies provided robust evidence supporting the efficacy and safety of HPV vaccines [12]. Similarly, other systematic reviews have concluded that the rates of serious adverse events for bivalent, quadrivalent, and nonavalent HPV vaccines are exceedingly low and do not significantly differ from those observed in placebo groups [37,38]. Building on these findings, data integrated from over 20 RCTs revealed no compelling evidence of increased risk among vaccinated individuals compared to their unvaccinated counterparts [13].

Central public health authorities—including the U.S. Centers for Disease Control and Prevention (CDC) and the European Medicines Agency (EMA)—have conducted ongoing post-marketing surveillance and have consistently found no statistically significant evidence linking HPV vaccines to severe adverse outcomes [16,39]. WHO’s Global Advisory Committee on Vaccine Safety (GACVS) has repeatedly affirmed the “very favorable safety profile” of HPV vaccines in publicly released statements [14]. Analyses of large-scale safety monitoring systems (e.g., Vaccine Adverse Event Reporting System (VAERS) and Vaccine Safety Datalink (VSD)) under the CDC’s purview have concluded that “no specific serious adverse events show an elevated incidence in HPV-vaccinated populations” [15]. Furthermore, WHO informational materials note that rare allergic reactions such as anaphylaxis occur at frequencies comparable to those associated with other routine immunizations [40].

A key focus of safety research involves the potential association between HPV vaccination and autoimmune or demyelinating disorders. A meta-analysis found no statistically significant increase in the incidence of autoimmune conditions following HPV vaccination [41]. A cohort study likewise confirmed that young females who received the quadrivalent vaccine did not exhibit an increased risk of autoimmune disorders [42]. In addition, a large-scale investigation conducted in two Nordic countries revealed no heightened risk for multiple sclerosis (MS) or related demyelinating diseases among HPV-vaccinated adolescents [43]. Collectively, these findings provide strong reassurance regarding the long-term safety of HPV immunization and effectively address concerns surrounding severe immune-mediated adverse events.

### 4.2. Specific Safety Concerns and Their Refutation

Since the initial rollout of HPV vaccines, concerns have been raised about chronic pain syndromes, autonomic dysfunction, and premature ovarian insufficiency, among other potential adverse outcomes. Nevertheless, global investigations have not produced conclusive evidence of a causal relationship between HPV vaccines and these alleged health issues.

#### 4.2.1. Chronic Pain Syndromes and Autonomic Dysregulation

Cases of complex regional pain syndrome (CRPS) and postural orthostatic tachycardia syndrome (POTS) were reported in specific populations in Denmark and Japan, prompting public and scientific speculation regarding a potential vaccine association [16,17]. However, these rare conditions can arise spontaneously in adolescence, suggesting that temporal coincidence with vaccination may have been overemphasized. In 2015, the EMA’s expert committee undertook a large-scale review and concluded that “no causal association exists between HPV vaccines and CRPS/POTS” [16]. A review by Kim similarly indicated that the proportion of CRPS reports relative to the total vaccinated population was exceedingly low, with no discernible nationwide upward trend [17].

#### 4.2.2. Autoimmune Disorders

Fears of an increased incidence of autoimmune disease escalation following HPV vaccination have been repeatedly tested in multiple cohort studies and meta-analyses, which have consistently reported no elevated risk [27,28]. Detailed comparisons of incidence rates in vaccinated versus unvaccinated groups showed no statistically significant differences, undercutting the hypothesis that HPV vaccines provoke novel or exacerbated autoimmune responses.

#### 4.2.3. Premature Ovarian Insufficiency (POI)

Allegations that HPV vaccination may trigger POI or otherwise compromise fertility also have surfaced in certain media accounts. However, large-scale analyses [15,40] have found no robust evidence of an increased risk of ovarian dysfunction. Reported instances appear to be sporadic and are most likely reflective of baseline occurrences unrelated to immunization. Notably, the U.S. CDC, which monitors the relationship between adolescence-targeted vaccines and POI, has detected no remarkable signal suggesting a greater likelihood of ovarian insufficiency attributable specifically to HPV vaccination [15].

### 4.3. Japan’s Domestic Investigations and Findings

Following the 2013 suspension of proactive HPV vaccine recommendations, a series of case reports emerged in Japan describing suspected adverse events following vaccination. In response, multiple investigative efforts have been undertaken to evaluate the validity of these claims.

#### 4.3.1. Expert Review Panels and Clinical Assessment

Since 2013, Japan’s Ministry of Health, Labour and Welfare has convened expert committees on vaccine side effects, gathering data from clinicians on suspected adverse cases [44]. However, the reported symptoms present a broad spectrum and often lack distinctive clinical markers, suggesting a possible association with orthostatic dysregulation or psychosomatic factors commonly manifest during adolescence. Ongoing reviews have yet to produce conclusive evidence of a direct causal relationship between the HPV vaccine and chronic pain or neurological impairment [18,25].

#### 4.3.2. Nagoya City’s Large-Scale Survey

Between 2015 and 2016, the City of Nagoya conducted a comprehensive survey involving approximately 30,000 participants to assess whether adverse symptoms were more prevalent among HPV vaccine recipients than non-recipients. Statistically significant differences were not observed, implying that symptom frequency was not significantly higher in vaccinated cohorts [19]. The findings were subsequently published in an international peer-reviewed journal and received considerable attention both domestically and internationally. Consequently, there is growing recognition that concerns about serious vaccine-related harm lack empirical support.

#### 4.3.3. Ongoing Safety Surveillance

Up to the official reinstatement of proactive recommendations in 2022, the Ministry of Health, Labour and Welfare and various professional societies regularly issued summaries of HPV vaccine safety data [25,44]. Because widespread vaccination had effectively ceased during the suspension period, the incidence of newly reported adverse events remained low, with no significant shift observed in the national safety profile [6,20,26]. In essence, neither Japanese nor international investigations have identified compelling evidence to suggest that HPV vaccines pose a substantial health risk.

## 5. Communication, Risk Perception, and Strategies for Recovery

### 5.1. Mismatch Between Evidence and Public Concern

Despite extensive scientific evidence affirming the high efficacy and favorable safety profile of HPV vaccines, vaccine hesitancy remains a significant issue in numerous countries. Scholars attribute this phenomenon to a pronounced gap between public risk perceptions and the available scientific data [8,45,46]. However, it is essential to acknowledge that the complete absence of risk cannot be scientifically demonstrated. While the public often desires “zero risk”, scientific assessments typically operate on margins and comparative probabilities. This discrepancy makes effective risk communication challenging, as people may perceive any reported adverse events, however rare, as evidence of potential danger rather than as part of a broader risk–benefit framework. In the case of HPV vaccines, the primary health benefits—namely, cancer prevention—emerge only in the long term. In contrast, apprehensions about immediate or severe adverse effects can loom disproportionately large [47]. Moreover, cognitive biases, such as omission bias (wherein risks of inaction are underestimated) and the availability heuristic (wherein striking anecdotes shape perceived risk), have been identified as key factors exacerbating HPV vaccine concerns [45,48].

A notable Canadian study by Feinberg et al. (2015) [49] analyzed over 3000 online comments responding to news articles about HPV vaccination, revealing a public discourse characterized by strong support and vocal opposition. Positive commenters emphasized vaccine safety and the benefits of cancer prevention. In contrast, negative sentiments often reflected distrust toward pharmaceutical companies, concerns about safety and efficacy, and debates around the appropriateness of school-based vaccination programs, particularly within religious contexts [49]. This analysis highlights how social media and news commentary platforms serve as important arenas where public attitudes are shaped and contested, reflecting complex societal and cultural dynamics.

Similarly, a recent nationwide French survey by Gauna et al. (2023) [50] illustrates the multifaceted nature of vaccine hesitancy. Their cluster analysis identified four distinct attitudinal groups—informed supporters, objectors, uninformed supporters, and uncertain individuals—each characterized by differing levels of knowledge, risk perception, and vaccine acceptance [50]. Notably, concerns about vaccine safety and side effects were prevalent even among the informed supporters, highlighting that vaccine hesitancy cannot be attributed solely to informational deficits but involves complex social, cultural, and psychological dimensions. These findings underscore the critical need for tailored communication strategies that address the specific concerns and contexts of diverse population subgroups to enhance HPV vaccine uptake effectively.

A notable Brazilian study by Minakawa and Frazão (2024) examined the print media narratives during the introduction of the HPV vaccine in Brazil, highlighting the media’s pivotal role in shaping public debate [51]. Their discourse analysis revealed that both proponents and opponents of the vaccine employed persuasive strategies centered around effectiveness, safety, and economic considerations. However, despite the wealth of scientific evidence supporting the vaccine, the media often blurred the lines between scientific facts and opinion, at times amplifying misinformation and creating a polarized public discourse. This phenomenon reflects the challenges faced in countries where media influence, misinformation, and public trust dynamics interact complexly to affect vaccine acceptance.

In Japan, media coverage of alleged adverse events, coupled with the government’s decision to halt proactive HPV vaccination recommendations, reinforced the perception that the vaccine harbored “some serious defect”, thereby undermining public trust [7,8]. Although multiple scientific reviews consistently concluded no significant safety concerns, these messages were often overshadowed by widespread anxiety, contributing to a sustained decline in vaccination uptake [20]. In addition, Nomura et al. (2024) comprehensively analyzed Japan’s complex sociocultural landscape regarding vaccine hesitancy, emphasizing how misinformation, historical distrust, and media dynamics collectively shape public perceptions and trust [52]. Their findings highlight the importance of culturally tailored communication strategies to address vaccine hesitancy in Japan.

In addition to these factors, past experiences with pharmaceutical controversies have contributed to lingering public skepticism. For example, as highlighted in the Canadian study by Feinberg et al. (2015), references to the Vioxx (rofecoxib) scandal—a nonsteroidal anti-inflammatory drug withdrawn due to safety concerns—frequently emerged in public discourse as a symbol of distrust toward pharmaceutical companies and regulatory bodies [49]. Such historical precedents amplify fears about vaccine safety, even when scientific evidence robustly supports immunization programs.

These intertwined issues of media influence, cognitive biases, and historical distrust underscore the complexity of addressing HPV vaccine hesitancy. Therefore, effective recovery strategies must extend beyond simply presenting scientific data. They require multifaceted, culturally sensitive communication approaches that engage communities, counter misinformation, and rebuild trust through transparency and dialog.

### 5.2. Lessons from Other Countries and Effective Interventions

#### 5.2.1. Denmark’s Approach to Restoring Coverage

Around 2013, Denmark experienced a notable surge in reports alleging HPV vaccine side effects, causing coverage rates to plummet from over 80% to approximately 40%. In response, Danish health authorities launched a large-scale communication campaign under the banner “Stop HPV (Stop HPV and stop cervical cancer)”, collaborating closely with healthcare providers, cancer advocacy groups, and media outlets to counter misinformation [10,11]. Their swift and transparent dissemination of accurate vaccine information, coupled with interactive outreach to parents and adolescents, is widely credited with the country’s relatively rapid recovery of vaccination rates.

#### 5.2.2. The Irish “Alliance” Model

A similar downturn was observed in Ireland around 2015, where negative discussions on social media drove HPV vaccination rates for middle- and high-school girls down from 87% to the vicinity of 50%. In response, the government partnered with professional associations, educators, and patient organizations to form the “HPV Vaccination Alliance”, mounting an intensive campaign to dispel social media rumors and alleviate public anxiety [53]. Personal stories of cervical cancer patients, who garnered substantial empathy through media appearances, further contributed to restoring public confidence. Strengthening the infrastructure for school-based vaccine delivery also proved instrumental in driving a notable rebound in vaccination coverage [11,53]. Collectively, these cases highlight the importance of promptly and transparently addressing public doubts and maintaining consistent messaging among multiple stakeholders to restore uptake within a short time frame.

### 5.3. Strategic Approaches for Japan

#### 5.3.1. Proactive Dissemination of Scientific Evidence and Clear Risk Communication

Due to Japan’s prolonged suspension of proactive HPV vaccination recommendations, the general public remains inadequately informed about the vaccine’s fundamental benefits and safety data [20]. Government agencies and healthcare professionals should aim to communicate information about HPV-associated cancer risks and the vaccine’s preventive efficacy in a clear, data-driven format—such as through numerical evidence and visual aids—while also emphasizing the rarity of adverse events and the predominantly mild nature of any side effects [46,47]. In addition, mirroring overseas examples where specialists actively engage with both traditional media and social media platforms to swiftly counteract misinformation may be essential [21,22].

#### 5.3.2. Interactive Communication and Enhanced Opportunities for Vaccination in Clinical Settings

Effective risk communication requires not only one-way information provision but also structured dialogs that directly address the concerns and uncertainties of parents and adolescents [45,48]. For instance, establishing dedicated vaccination opportunities at pediatric or obstetric clinics and incorporating HPV vaccine information into routine adolescent health checks can normalize HPV vaccination as a standard preventive measure, raising uptake rates [54].

Although school-based vaccination programs are proven to boost coverage in many countries, Japan’s legal and logistical environment often complicates mass vaccination in educational settings. Therefore, interventions such as group health education, on-site informational sessions, and direct engagement with parents and students—akin to the Danish model—could offer vital pathways for building trust.

Recently, a grassroots initiative in Japan known as “*Minpapi*” has garnered attention for its citizen-led approach. This project involves healthcare providers, researchers, parents, and young adults, aiming to disseminate accurate, accessible HPV vaccine information through diverse channels such as social media and in-person events [55,56]. As noted by Takahashi et al. [55], gaps in knowledge between parents and adolescents pose a significant hurdle to HPV vaccination; *Minpapi*’s community-based model seeks to bridge this divide by facilitating mutual learning and collective problem-solving. Through collaboration among healthcare systems, schools, government agencies, and civic groups, a more interactive form of communication can develop, potentially promoting greater acceptance and trust. Preliminary reports from *Minpapi* indicate that its workshops and online engagement have reached individuals who had not previously considered vaccination [49,50]. If scaled up, such bottom-up educational initiatives could complement clinical and institutional policies, contributing meaningfully to the restoration of HPV vaccination coverage in Japan.

#### 5.3.3. Prioritizing Catch-Up Interventions for Missed Cohorts

One of Japan’s most pressing challenges is assisting the so-called “lost generation” of individuals who missed vaccination opportunities between 2013 and 2021. Although the Ministry of Health, Labour and Welfare has introduced a time-limited “catch-up vaccination” program, critics note that awareness of this initiative remains limited, especially among women in their early twenties or older who may face logistical barriers, such as inflexible work schedules [6,26]. Targeted outreach strategies, including direct notifications from local governments, partnerships with universities and workplaces, and potentially extending public funding beyond the current deadline, may improve program uptake.

Furthermore, adult women who have remained unvaccinated could benefit from integrated strategies that link HPV immunization with cervical cancer screening, thus reducing their lifetime risk [57]. Strengthening secondary prevention while concurrently boosting adolescent vaccination rates can yield a substantial reduction in cervical cancer incidence and mortality over the long term.

### 5.4. What Does This Review Add?

#### 5.4.1. Q1: What Are the Unique Features of HPV Vaccine Hesitancy in Japan, and How Does This Review Clarify Them?

A1: This review integrates recent domestic and international evidence to demonstrate that HPV vaccine hesitancy in Japan has been driven by the prolonged governmental suspension of recommendations, intense media sensationalism, and limited direct communication from healthcare professionals. The persistence and magnitude of these factors are distinct when compared to other high-income countries, where vaccine confidence recovered more rapidly.

#### 5.4.2. Q2: What Evidence-Based Strategies for Restoring HPV Vaccine Uptake Are Highlighted in This Review?

A2: Drawing on lessons from Denmark and Ireland, the review highlights that transparent, interactive risk communication, consistent messaging from trusted stakeholders, and bottom-up educational initiatives are critical for rebuilding public trust. The review further evaluates the adaptability of these strategies to Japan’s specific sociocultural context.

#### 5.4.3. Q3: How Does This Review Contribute to Future Policy and Research Directions?

A3: This review provides a comprehensive synthesis of policy, communication, and societal factors contributing to vaccine hesitancy, offering actionable recommendations for policymakers and practitioners. It also identifies gaps in the current Japanese response and proposes priorities for future research and targeted interventions to close the coverage gap and reduce the burden of cervical cancer.

## 6. Conclusions

The HPV vaccine plays an essential role in preventing HPV-associated diseases, including cervical cancer, and has garnered widespread recognition for its efficacy and safety across numerous countries. As this review illustrates, Japan’s abrupt suspension of proactive vaccine recommendations in 2013 led to a dramatic decline in vaccination rates and ushered in a prolonged “vaccination crisis”. Nonetheless, recent measures to reinstate recommendations and implement catch-up programs appear to be initiating a slow but discernible recovery. Drawing on the successes observed in other nations, there is reason to believe that rigorous disclosure of scientific evidence, strengthened collaboration among healthcare providers, governmental agencies, and the educational sector, as well as effective two-way communication with parents and adolescents, can feasibly restore HPV vaccine coverage to previously high levels [10,11,58,59,60].

Multiple factors contributed to Japan’s prolonged low uptake of HPV vaccination, including an overemphasis on suspected side effects in media reporting and on social media, inadequate risk communication on the part of public authorities, and sociocultural environments that discouraged more proactive engagement by healthcare professionals [7,9,36,61,62]. Consequently, simply presenting scientific data is insufficient; instead, a supportive environment that swiftly addresses misinformation and assists individuals in making informed decisions is crucial. Specific measures include conveying the balance between potential future cancer risks and the relatively minor risk of side effects, fostering “dialog-based” communication to address questions and anxieties, and establishing broader access points in schools, universities, and clinical settings such as obstetrics, gynecology, and pediatrics departments [54,55,56,63,64,65,66].

Moreover, concerted action is needed to broaden the “catch-up vaccination” initiatives targeting the so-called “lost generation” of individuals who missed vaccination opportunities after 2013, emphasizing equity and public health. For those now entering adulthood, strategies involving social media campaigns, workplace health examinations, and university partnerships could be key to reaching populations with limited time or information [26,67,68]. Additionally, integrating HPV vaccination with cervical cancer screening could help reduce the cumulative disease burden [35,57].

Moving forward, Japan must take into account global trends, including the transition to nonavalent HPV vaccines and the expansion of vaccination programs to include male adolescents. Such changes should be incorporated into comprehensive national strategies that are sensitive to Japan’s unique cultural and social context. By leveraging robust scientific evidence and learning from international experience, Japan can mitigate ongoing concerns about HPV vaccination and substantially reduce the future burden of HPV-related cancers, including cervical cancer.

## Data Availability

No new data were generated or analyzed in this study. Data sharing is not applicable.

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
