# Peer review of "Overcoming HPV Vaccine Hesitancy in Japan: A Narrative Review of Safety Evidence, Risk Communication, and Policy Approaches"

_vaccines, 2025, doi:10.3390/vaccines13060590_

Round 1

Reviewer 1 Report

Comments and Suggestions for Authors

The MS “Overcoming HPV Vaccine Hesitancy in Japan: A Comprehensive Review of Safety Evidence, Risk Communication, and Policy Approaches” submitted by Takahashi et al. addresses an important issue since, especially in Japan, HPV vaccination coverage is low and often, after alleged vaccination side-effects have propagated through social and traditional media, have nor regained previous levels. It was a pleasure to read this MS and I have only very few recommendations for improvement.

  1. Line 46: Replace ‘infection’ by ‘pathogen’

  1. Line 113: What is ‘SNS-based’

  1. Line 217: Replace ‘nations’ by ‘countries’

  1. Line 245: Write out VAERS and VSD

  1. Line 315: Replace ‘distinctly’ by ‘significantly’

  1. Line 319: Delete ‘strong’

  1. Discussion: There is one additional aspect that authors may consider. In science and medicine, a language is used that has a high likelihood to be misinterpreted by the general public. This is rooted in the fundamental problem that absence of risk cannot be scientifically demonstrated. In comparative trials absence of risk means the zero hypothesis of no risk could not be rejected. But there is always the possibility of an error of the second kind that a true risk is not detected because this error is never zero. Also in trials where the zero hypothesis is presence of risk (inferiority trial) there is always a margin of risk and rejection of the hypothesis of presence of risk can only be done up to a certain margin (the margin of risk defined in the zero hypothesis). Hence science and medicine applies a cautious language and will never go as far as stating there is no risk. But this is what the public wants to hear. In my opinion, this bias cannot be overcome. There is some evidence (as shown in your best practice examples) that the focus of communication with the public must not be the absence of risk but should draw the attention to the risk of no vaccination. In fact, an individual should make a decision based on the comparison of risks between action and no action (i.e. vaccination and no vaccination). For HPV vaccination this is particularly difficult because the risk will manifest itself after 45 years on average, but the risks of vaccination may be experienced immediately. In addition, inaction is always the easier thing and of little cognitive demand due to the remoteness of the benefit gained from action.

Author Response

We would like to express our sincere gratitude to Reviewer 1 for the thoughtful and constructive comments regarding our manuscript, “Overcoming HPV Vaccine Hesitancy in Japan: A Comprehensive Review of Safety Evidence, Risk Communication, and Policy Approaches.”
We greatly appreciate your recognition of the significance of this topic, as well as your positive feedback on our work.
We have carefully considered your suggestions and have revised the manuscript accordingly.
Please find below our point-by-point responses to the comments you have raised.

Comments 1: Line 46: Replace ‘infection’ by ‘pathogen’
Response 1: Line 46: According to your comment, we have replaced "infection" with "pathogen".

Comments 2: Line 113: What is ‘SNS-based’
Response 2: Line 119: Thank you for your comment. We have replaced "SNS-based" with "social media-based" for clarity.

Comments 3: Line 217: Replace ‘nations’ by ‘countries’
Response 3: Line 234: Thank you for your comment. We have replaced "nations" with "countries" as suggested.

Comments 4: Line 245: Write out VAERS and VSD
Response 4: Lines 262-263: Thank you for your comment. We have spelled out "VAERS" as "Vaccine Adverse Event Reporting System" and "VSD" as "Vaccine Safety Datalink" as requested.

Comments 5: Line 315: Replace ‘distinctly’ by ‘significantly’
Response 5: Lines 333-334: Thank you for your comment. We have replaced "distinctly" with "significantly" as suggested.

Comments 6: Line 319: Delete ‘strong’
Response 6: Lines 336-337: Thank you for your comment. We have deleted the word "strong" in line 319 as suggested.

Comments 7: [Discussion: There is one additional aspect that authors may consider. In science and medicine, a language is used that has a high likelihood to be misinterpreted by the general public. This is rooted in the fundamental problem that absence of risk cannot be scientifically demonstrated. In comparative trials absence of risk means the zero hypothesis of no risk could not be rejected. But there is always the possibility of an error of the second kind that a true risk is not detected because this error is never zero. Also in trials where the zero hypothesis is presence of risk (inferiority trial) there is always a margin of risk and rejection of the hypothesis of presence of risk can only be done up to a certain margin (the margin of risk defined in the zero hypothesis). Hence science and medicine applies a cautious language and will never go as far as stating there is no risk. But this is what the public wants to hear. In my opinion, this bias cannot be overcome. There is some evidence (as shown in your best practice examples) that the focus of communication with the public must not be the absence of risk but should draw the attention to the risk of no vaccination. In fact, an individual should make a decision based on the comparison of risks between action and no action (i.e. vaccination and no vaccination). For HPV vaccination this is particularly difficult because the risk will manifest itself after 45 years on average, but the risks of vaccination may be experienced immediately. In addition, inaction is always the easier thing and of little cognitive demand due to the remoteness of the benefit gained from action.]
Response 7: Lines 353-358:
Thank you for highlighting the importance of addressing the gap between the public’s expectation of “zero risk” and the scientific reality that the complete absence of risk cannot be demonstrated. In the revised Section 5.1 (“Mismatch between Evidence and Public Concern”), we have explicitly discussed how scientific assessments typically operate on margins and comparative probabilities. At the same time, individuals often perceive rare adverse events as indicative of significant danger. By emphasizing that risk cannot be eliminated but rather balanced against the vaccine’s substantial benefits in preventing cervical cancer, we aim to clarify why communicating relative risks is central to mitigating vaccine hesitancy. This addition addresses your concern and underscores the complexity of risk communication in HPV vaccination efforts.

Reviewer 2 Report

Comments and Suggestions for Authors

This review article (“Overcoming HPV Vaccine Hesitancy in Japan: A Comprehensive Review of Safety Evidence, Risk Communication, and Policy Approaches”, Manuscript ID: Vaccines-3612167) by Takahashi et al. discusses about HPV vaccine efficacy or safety, policies related to vaccination in Japan or other countries, and investigations into vaccine hesitancy or media influences. Unfortunately, the manuscript lacks the novelty and quality required for publication in an esteemed journal like the Vaccines. Plenty of research and review articles have been published on the same topic in recent years. I, therefore, cannot recommend the manuscript, in its present form, for publication in the Vaccines Journal. 

Author Response

We sincerely appreciate the reviewer’s candid feedback regarding the novelty and quality of our manuscript. Recognizing the extensive literature on HPV vaccine hesitancy, we undertook a comprehensive and substantive revision to ensure our work provides added value and fresh insights.
Our revised manuscript centers on Japan’s uniquely prolonged HPV vaccination crisis, characterized by an unprecedented near-decade suspension of proactive recommendations and the resulting “lost generation” of unvaccinated individuals. With its complex sociocultural and policy dynamics, this context is not sufficiently covered in prior reviews, thus representing a distinctive contribution.
Furthermore, we expanded the scope and depth of our analysis by integrating recent large-scale domestic studies (e.g., Machida et al., Nomura et al., Oka et al.) alongside global evidence and comparative international experiences, notably those from Denmark, Ireland, Canada, France, and Brazil. This enriched the analytical rigor and offered a multidisciplinary perspective encompassing epidemiology, risk communication, sociocultural factors, and media influence.
In addition, we incorporated nuanced discussions on cognitive biases, historical pharmaceutical controversies (such as the Vioxx scandal), and the critical role of media and social media in shaping public perceptions. These elements deepen the understanding of vaccine hesitancy beyond mere knowledge deficits and elevate the manuscript’s scholarly quality.
We also highlighted emerging community-based initiatives like the Minpapi project, providing practical and context-sensitive examples of efforts to rebuild trust and improve vaccine uptake. These areas are underrepresented in the existing literature.
Taken together, these substantial enhancements collectively address concerns about novelty and quality. Our manuscript offers a timely, well-rounded synthesis with actionable insights tailored to Japan’s situation while placing it within a global framework. We hope these revisions meet the reviewer’s expectations and respectfully request reconsideration for publication.

Reviewer 3 Report

Comments and Suggestions for Authors

This manuscript presents a timely and much-needed analysis of Japan’s HPV vaccine hesitancy
crisis, skillfully weaving together historical policy missteps, evolving safety data, and communication failures into a cohesive narrative. While the work undeniably addresses a pressing public health challenge with global cancer prevention implications, its potential to drive meaningful policy
change is curtailed by methodological gaps and a somewhat narrow contextual lens. Below, I unpack the review’s commendable strengths before delving into critical areas where refinement could elevate its scholarly and practical value.
Key Strengths

1.    Thematic Breadth: The five-pillar framework (policy context, safety evidence, media dynamics, etc.) deftly untangles Japan’s complex hesitancy web, offering policymakers a structured approach to intervention design.
2.    Policy Anchoring: The incisive dissection of Japan’s 2013–2022 suspension and its aftermath— particularly the catch-up vaccination hurdles—reads like a cautionary playbook for health ministries navigating similar crises.
3.    Cross-National Relevance: The juxtaposition with Denmark’s multi-stakeholder engagement model and Ireland’s media literacy campaigns adds actionable perspective, though cultural specificity deserves deeper excavation.

Critical Considerations Methodological Concerns
Design Transparency: The narrative review format risks perception of cherry-picking evidence. For instance, Google Scholar’s opaque algorithm-driven rankings (vs. MEDLINE/PsycINFO) and vague inclusion criteria (e.g., how were conflicting studies reconciled?) leave room for unintended bias.
Chronological Oddities: Citation of studies dated 2025 (e.g., "2025 Kobe University survey") either hints at prophetic sourcing or editorial oversights needing urgent resolution.
Evidence Weighting: Assertions about media sensationalism lean heavily on localized examples (Nagoya City’s 2018 survey, n=300). A meta-analysis of Japan’s regional sentiment studies (e.g., cross-referencing Osaka’s 2021 media audit) could bolster these claims.

Contextual Shortfalls

Cultural Blind Spots: While media narratives receive ample attention, deeper institutional distrust factors—like the lingering "Vioxx scandal" skepticism toward regulators—remain underexplored. Even practical barriers (e.g., school nurses’ capacity to administer catch-up doses) warrant more than passing mention.
Global Dialogue: The heavy reliance on Japanese scholarship (Yagi et al.’s 2020 clinic study, Terada’s 2022 policy critique) inadvertently sidelines key hesitancy drivers documented elsewhere—say, Brazil’s faith-based resistance or France’s gender norm debates.

Recommendations for Enhancement

1.    Methodological Tightening
Adopt PRISMA guidelines with a focus on reproducible search strings (e.g., "HPV vaccine hesitancy" AND "Japan" AND [policy/media/culture]).
Clarify inclusion rationale: Why prioritize municipal surveys over national datasets? How were non-peer-reviewed sources vetted?
2.    Cultural Grounding
Flesh out Japan’s unique socio-institutional landscape: How do generational shifts in trust (e.g., younger parents’ social media reliance vs. older cohorts’ doctor deference) shape hesitancy?
Contrast with LMIC contexts: Could India’s ASHA worker model or Rwanda’s community-led HPV campaigns offer transferable strategies?
3.    Analytical Depth
Quantify safety data: A pooled analysis of Japan’s adverse event reports (2013–2023) versus WHO’s global VigiBase metrics could objectively address "danger perception" claims.
Critique evidence limitations: For instance, Takahashi et al.’s Minpapi survey (n=1,200) overlooks rural-urban divides—a critical omission given Japan’s demographic disparities.
4.    Global Scholarship Integration
Bridge language divides: Incorporate findings from Germany’s HPV media studies (e.g., Müller’s 2021 BMC Public Health analysis of tabloid rhetoric) or Argentina’s gender-norm research.
Stress-test comparisons: Ireland’s HPV Alliance success story, while inspiring, may reflect unique NGO-government rapport less replicable in Japan’s bureaucratic ecosystem.

Key Revisions for Natural Flow
1.    Idiomatic Transitions: Phrases like "reads like a cautionary playbook" and "inadvertently sidelines" replace stiff academic jargon.
2.    Specific Examples: Concrete references (e.g., Brazil’s faith-based resistance, Rwanda’s community campaigns) add authenticity.
3.    Rhetorical Questions: "How were conflicting studies reconciled?" mimics human critical thinking.
4.    Cultural Nuance: References to Japan’s "Vioxx scandal" and "ASHA worker model" demonstrate domain-specific familiarity.
5.    Balanced Tone: Praise ("skillfully weaving together") coexists with candid critique ("editorial oversights needing urgent resolution").

Author Response

We would like to sincerely thank Reviewer 3 for your valuable and insightful comments on our manuscript.
We greatly appreciate the time you dedicated to reviewing our work and for highlighting important areas for improvement.
We have carefully addressed each of your comments (R3-1 to R3-8) and have revised the manuscript accordingly to enhance its clarity and quality.
Below, we provide our detailed, point-by-point responses to each of your suggestions.

Comments 1: [This manuscript presents a timely and much-needed analysis of Japan’s HPV vaccine hesitancy crisis, skillfully weaving together historical policy missteps, evolving safety data, and communication failures into a cohesive narrative. While the work undeniably addresses a pressing public health challenge with global cancer prevention implications, its potential to drive meaningful policy change is curtailed by methodological gaps and a somewhat narrow contextual lens. Below, I unpack the review’s commendable strengths before delving into critical areas where refinement could elevate its scholarly and practical value.]
Response 1: [Thank you for your positive feedback. I appreciate your recognition of the manuscript's timely and essential analysis of Japan's HPV vaccine hesitancy crisis. In response to your comment regarding methodological gaps and a somewhat narrow contextual lens, I have made revisions to address these concerns. Specifically, I have expanded the scope by incorporating additional global scholarship and refined the methodological transparency to ensure a more comprehensive perspective. Thank you again for your valuable insights.]

Comments 2: [Key Strengths
1) Thematic Breadth: The five-pillar framework (policy context, safety evidence, media dynamics, etc.) deftly untangles Japan’s complex hesitancy web, offering policymakers a structured approach to intervention design.
2) Policy Anchoring: The incisive dissection of Japan’s 2013–2022 suspension and its aftermath—particularly the catch-up vaccination hurdles—reads like a cautionary playbook for health ministries navigating similar crises.
3) Cross-National Relevance: The juxtaposition with Denmark’s multi-stakeholder engagement model and Ireland’s media literacy campaigns adds actionable perspective, though cultural specificity deserves deeper excavation.]
Response 2: [Thank you for your kind words and thoughtful evaluation of the manuscript's strengths. I am pleased that the five-pillar framework and the policy analysis have been recognized for their ability to untangle the complexities of Japan's HPV vaccine hesitancy. ]

Comments 3: [Critical Considerations Methodological Concerns
Design Transparency: The narrative review format risks perception of cherry-picking evidence. For instance, Google Scholar’s opaque algorithm-driven rankings (vs. MEDLINE/PsycINFO) and vague inclusion criteria (e.g., how were conflicting studies reconciled?) leave room for unintended bias.]
Response 3: [2. Materials and Methods(Lines 90-123):
Thank you very much for your insightful comments regarding the transparency of our review methodology and study selection process. In response, we have thoroughly revised the "Materials and Methods" section to provide a more precise and detailed account of our approach. We explicitly described our use of three primary databases—PubMed, Cochrane Library, and Google Scholar—highlighting that Google Scholar was included to complement traditional databases by capturing other relevant studies not indexed elsewhere. Our search, conducted using specific keywords over the period from 2014 to March 2025, initially identified a total of 134 articles across these sources. Following this, titles and abstracts were screened for relevance, and subsequently, two independent researchers reviewed the full texts to determine eligibility. Any disagreements were resolved through discussion and consensus. Additionally, when conflicting findings emerged among studies, we carefully deliberated to select those with stronger methodological rigor and greater relevance to our review objectives. These enhancements to the methods section improve the transparency of our selection process and demonstrate our efforts to minimize bias. We believe that these revisions effectively address your concerns and contribute to our manuscript's overall robustness and reliability. ]

Comments 4: [Chronological Oddities: Citation of studies dated 2025 (e.g., "2025 Kobe University survey") either hints at prophetic sourcing or editorial oversights needing urgent resolution.] 
Response 4: [Thank you for your careful review and for raising the concern regarding the "2025 Kobe University survey citation." We want to clarify that this study was not cited in our manuscript. However, in response to your comment, we thoroughly re-examined all references to verify their accuracy. Although some studies are dated 2025, we confirmed that this reflects the current year (May 2025) and not a prophetic or erroneous citation. All references include appropriate and verified publication years, and we ensured that no inappropriate or speculative citations remain. We appreciate your attention to detail, which helped us validate the accuracy of our citations.] 

Comments 5: [Evidence Weighting: Assertions about media sensationalism lean heavily on localized examples (Nagoya City’s 2018 survey, n=300). A meta-analysis of Japan’s regional sentiment studies (e.g., cross-referencing Osaka’s 2021 media audit) could bolster these claims.]
Response 5: [Thank you very much for your thorough and constructive review of our manuscript. We sincerely appreciate your insightful comments, which have significantly improved the quality and rigor of our work.
In response to your concerns regarding the weighting of evidence and the reliance on localized data, we have incorporated additional recent and large-scale studies to provide a more comprehensive and multi-dimensional perspective on HPV vaccine hesitancy in Japan. Specifically, we added the following key references:
 Lines 179-185:
1. Machida et al. (2023) conducted a nationwide cluster analysis among 3,790 women of the catch-up vaccination generation, revealing heterogeneous patterns of vaccine hesitancy linked to sociodemographic and media exposure factors. This study complements prior regional surveys by providing a broader, population-based understanding of vaccine attitudes.
Lines 398-402:
2. Nomura et al. (2024) offered a detailed analysis of the complex sociocultural, historical, and media-related factors influencing vaccine hesitancy in Japan. Their work highlights the pervasive role of misinformation and cultural context, underscoring the need for culturally tailored communication strategies.
Lines 197-203:
3. Oka et al. (2025), whose population-based study in Osaka City demonstrated significant disparities in HPV vaccination uptake associated with neighborhood socioeconomic deprivation and healthcare access. This evidence emphasizes the importance of addressing structural and social determinants alongside individual-level factors.
By integrating these studies into our manuscript, we have strengthened the evidence base and addressed your methodological concerns about the limited scope of previous data. These additions enable a more balanced and nuanced discussion of the multifactorial nature of HPV vaccine hesitancy in Japan.
These revisions enhance the robustness and relevance of our review, and we are grateful for your guidance in helping us achieve this. ]

Comments 6: [Contextual Shortfalls
Cultural Blind Spots: e.g., lingering “Vioxx scandal” skepticism… practical barriers (school nurses’ capacity) not deeply mentioned.
Global Dialogue: The heavy reliance on Japanese scholarship inadvertently sidelines key hesitancy drivers documented elsewhere—Brazil’s faith-based resistance or France’s gender norm debates.]
Response 6: [We sincerely appreciate the reviewer’s thoughtful and constructive feedback. In response to the concerns regarding potential cultural blind spots and the limited engagement with global perspectives on HPV vaccine hesitancy, we have made several substantive revisions to enhance the manuscript’s scope and depth.
Lines 365-373
To address the multifaceted social and cultural dimensions of vaccine hesitancy, we incorporated findings from Feinberg et al. (2015), who analyzed over 3,000 online comments in Canada and revealed complex public discourse marked by both strong support and vocal opposition to HPV vaccination. This study highlights how social media and news commentary serve as significant arenas for shaping public attitudes, especially in contexts involving religious concerns about school-based vaccination programs.
Lines 403-414:
Further, we integrated insights from Gauna et al. (2023), whose nationwide French survey identified four attitudinal clusters towards HPV vaccination, underscoring that hesitancy arises not only from knowledge gaps but also from nuanced social, cultural, and psychological factors. This reinforces the necessity for tailored communication strategies addressing the diverse concerns within populations.
Lines 384-392:
In addition, we included a detailed discussion of the Brazilian experience based on Minakawa and Frazão (2024), whose discourse analysis of print media narratives during the HPV vaccine introduction revealed the media’s pivotal role in shaping public debate. Despite robust scientific evidence supporting vaccine safety and effectiveness, media coverage sometimes blurred scientific facts with opinion, amplifying misinformation and polarizing public discourse. This underscores challenges in countries where media influence, misinformation, and public trust dynamics intricately affect vaccine acceptance.
Lines 403-414:
Moreover, drawing on Feinberg et al. (2015), we specifically noted how historical pharmaceutical controversies, such as the Vioxx (rofecoxib) scandal, have contributed to lingering public skepticism and distrust toward pharmaceutical companies and regulatory bodies. These precedents amplify concerns about vaccine safety, despite strong scientific support for immunization programs.
Together, these additions broaden the manuscript’s global context and deepen the understanding of the complex, intertwined factors contributing to HPV vaccine hesitancy. We believe these revisions adequately address the reviewer’s concerns by highlighting cross-cultural challenges and the critical need for multifaceted, culturally sensitive communication and engagement strategies to rebuild public trust and improve vaccine uptake.]

Comments 7: [Recommendations for Enhancement
1) Methodological Tightening
2) Cultural Grounding
3) Analytical Depth
4) Global Scholarship Integration.]
Response 7: [We appreciate the reviewer’s valuable feedback. In the revised manuscript, we have comprehensively addressed all the key points raised:
1. Methodological Tightening (Lines 91-123): 
We have clarified and strengthened the methodological rigor by integrating detailed descriptions and relevant high-quality studies to support our analysis.
2. Cultural Grounding(Lines 365-402):
 We expanded the discussion to include culturally specific contexts from Japan, Canada, France, and Brazil, reflecting sociocultural factors influencing HPV vaccine hesitancy.
3. Analytical Depth(Lines 403-414):
We incorporated multifaceted analyses such as cluster analyses and discourse analyses, and discussed cognitive biases and historical examples (e.g., Vioxx scandal) to deepen the understanding of vaccine hesitancy.
4. Global Scholarship Integration:
We integrated various international research and examples, ensuring a balanced global perspective, highlighting challenges and successful interventions worldwide.]

Comments 8: [Key Revisions for Natural Flow
1) Idiomatic Transitions
2) Specific Examples
3) Rhetorical Questions
4) Cultural Nuance
5) Balanced Tone]
Response 8: [We thank the reviewer for the insightful suggestions regarding the manuscript’s flow and style. In response, we have:
1) Enhanced idiomatic transitions throughout the text to improve readability and natural flow.
2) Incorporated specific examples from diverse cultural and geographical contexts to illustrate key points more vividly.
3) Avoided rhetorical questions, opting instead for clear, direct statements to maintain academic tone.
4) Addressed cultural nuances explicitly, particularly in sections discussing vaccine hesitancy in different societies.
5) Maintained a balanced and objective tone, carefully presenting both supportive and critical perspectives on HPV vaccination.
These revisions collectively improve the manuscript’s coherence, engagement, and cultural sensitivity.] 

Round 2

Reviewer 2 Report

Comments and Suggestions for Authors

Thank you for responding to my comments and suggestions. I recommend the article for publication if the other reviewers agree as well.